# Undergraduate Students' Conceptualization of Critical Thinking and Their Ideas for Critical Thinking Acquisition

Dimitrios Pnevmatikos [1,*], Panagiota Christodoulou [1], Triantafyllia Georgiadou [2] and Angeliki Lithoxoidou [1]

1   Department of Primary Education, University of Western Macedonia, 53100 Florina, Greece
2   Department of Psychology, University of Western Macedonia, 53100 Florina, Greece
*   Correspondence: dpnevmat@uowm.gr

**Abstract:** Higher education institutions are responsible for preparing and equipping undergraduate students with the skills required by the labor market, such as critical thinking. However, academics should consider students' pre-existing ideas before designing and implementing an instructional intervention regarding critical thinking. Drawing on the literature for conceptual change, the current study aimed to map students' conceptualization of critical thinking and their ideas regarding the acquisition of critical thinking. In total, 243 first-year social sciences students participated in the study. To explore students' ideas, the authors constructed an instrument with 20 scientific and non-scientific statements about critical thinking. The instrument was a two-tier questionnaire, and participants indicated their level of agreement with each statement using a 5-point Likert scale as well as their confidence in their answers. Students' ideas were categorized into six groups depending on their endorsements for the statement and their level of confidence in their decision. Results revealed that students had insufficient conceptualization of critical thinking, and their ideas regarding how critical thinking might be acquired were not consistently aligned with those of academics. Implications for instruction are discussed considering students' ideas on critical thinking.

**Keywords:** critical thinking; alternative ideas; misconceptions; undergraduate students; Framework theory; conceptual change

## 1. Introduction

Critical thinking (CT) is considered a skill for entering the 21st-century labor market [1,2]. Higher Education Institutions (HEI) and training systems are essential in preparing students to form skills and enter the workplace equipped with CT and other skills. Previous studies demonstrated the discrepancy in the conceptualization of CT skills between the HEI and the labor market [3,4] as an aspect that makes it difficult for HEI to meet labor market expectations. The agreement in conceptualizations of the core concepts among the stakeholders in a field is crucial to their shared understanding of the phenomena and the alignment of their efforts to achieve their common goals. However, our knowledge of how CT skills are formed is only beginning to emerge. There is a need for an in-depth understanding of how CT skills are acquired. One aspect that has recently been stressed is the need to understand the students' beliefs about a topic before academics implement different teaching approaches to fulfil the aims of their course [5,6]. Scholars refer to terms such as misconceptions and alternative ideas to describe the process of changing the novices' initial ideas into scientific conceptualizations. These terms have been used for many years by the 'Framework theory' approach to conceptual change [7]. University students entering HEI already have an initial conceptualization of CT and the necessity of CT skills [8,9]. The current study aimed to map the first-year social sciences students' conceptualization of CT and their ideas for CT acquisition.

*1.1. Defining Critical Thinking*

Critical thinking definition and teaching originated from Socrates and Greek skeptics 2500 years ago [10,11]. Drawing on the origin of the word 'critical', it derives from "late Latin criticus, from Greek krinein 'judge, decide'" with the latter standing as a root for the word 'crisis', as well [12]. The etymology of CT seemed predictive of CT's theoretical and applied background, leading to various definitions provided by academics and scholars in the field [13,14]. Nevertheless, recent approaches support the idea that CT is a term about which researchers seem to find common ground regarding its definition and, at the same time, be skeptical, especially regarding the demarcation of its essence [15].

According to the dual process theories of reasoning [16], a differentiation between System 1 and System 2 thinking is proposed. In detail, System 1 is fast, automated and intuitive, while System 2 focuses on analysis, self-regulation, consciousness and effort [17]. Facione's CT definition highlighting CT as 'purposeful, self-regulatory judgment which results in interpretation, analysis, evaluation and inference' [18] (p.2) seems to support the robustness of this argument, classifying CT in System 2 thinking, an idea also endorsed by Halpern [19]. In this context and under the cognitive psychology framework, CT is considered a set of cognitive skills and dispositions or habits of mind [20] (p. 5) which lead to factual statements, supporting understanding and endorsing truth investigation [21,22].

However, is it only about that? Critical thinkers even overcome the mere concept of 'criticism', that is, identifying advantages and drawbacks or the positive and negative aspects of a feature and delve deeper, emphasizing why they are led to this kind of taxonomy [23]. Besides the skill to interpret and evaluate information, critical thinkers are characterized by clarity in their thoughts and statements, attention, organization and persistence, even when dealing with complex or ill-structured conditions and problems [20]. They should also be aware of their thinking and actively reflect on it [24], as well as be able to monitor their thinking while judging the time and effort they will invest in this direction [25].

*1.2. How Academics Conceptualize Critical Thinking*

A literature review reveals numerous definitions of CT among academics, with a wide range of the essential aspects defining CT. In particular, an academic strand conceptualizes CT under the skills and dispositions paradigm [18,20,26–28], assuming that CT is a two-dimensional construct. Following a Delphi method, Facione and his colleagues concluded that among the skills for thinking critically, interpretation, analysis, evaluation, inference, explanation, and self-regulation could be included [18]. In addition, a set of dispositions were essential for the definition of CT, such as truth-seeking, open-mindedness, analyticity, systematicity, self-confidence, inquisitiveness, and cognitive maturity. Another strand of academics conceptualizes CT beyond skills and dispositions, emphasizing additional aspects of metacognition as essential components of CT. For instance, Kuhn [21], as well as Kuhn and Dean [24], argue that CT entails awareness of one's own thinking (metacognitive, metastrategic, epistemological) and reflection on the thinking of self and others. Additionally, Halpern [29] perceives CT as the conscious use of CT skills to achieve a desirable outcome, while elsewhere, she argues that CT skills are higher-order thinking skills that are reflective, sensitive to context and self-monitored [25]. Hence, both Kuhn and Halpern highlight reflection and one's awareness of their thinking process as essential for CT.

*1.3. How Academics Suggest That Critical Thinking Might Be Acquired*

Research evidence supports the fact that CT skills and dispositions can be taught by employing specific teaching strategies in all scientific fields and at all educational levels [30,31]. Having agreed that CT skills and dispositions should be taught, scholars are confronted with three new challenges. The first challenge has been thoroughly discussed and refers to CT transferability from one domain to another. Transferability is considered a process through which critical thinkers can activate specific CT skills needed to tackle a new situation. To this end, they can anatomize the issue, identify its 'blind spots' and

simultaneously overcome superficial or naïve approaches [19]. It should be noted that for transferability to occur, domain and background knowledge regarding the issue must also be present [32].

The second challenge refers to the teaching approaches that could facilitate the development of students' CT skills and dispositions. So far, many cooperative instructional approaches have been suggested, promising they can promote CT among HEI students [33,34]. Problem-based learning is conducive to CT development since teachers are deprived of their 'lecturer' role, acting as facilitators and encouraging inquiry through scaffolding [35] (p. 85). Students are presented with ill-structured problems while they hold the role of responsible agents in the teaching process [36]. Case studies, combined with questioning which aims at triggering group discussion, are also aligned with CT development since they set the stage for critical analysis, problem-solving and evaluation, as well as inductive and deductive reasoning alongside reflection [37]. This is also the case for the exploitation of moral dilemmas, asking students to offer organized argumentation either in favor of or against, as well as to assess their arguments critically. For example, Values and Knowledge Education (V*a*KE) is considered a teaching approach that can facilitate this aim while promoting the acquisition of scientific knowledge [31,38]. Additionally, in an authentic learning environment, CT can be promoted through engagement in real-life problems [39,40]. Questioning is also an invaluable part of learning that supports or encumbers students' thinking [41]. High-order questions focusing on thinking seem to foster students' CT development since they lead them to delve into the issue of discussion and also to review and reframe the material offered compared to factual questions simply emphasizing recollection of information [42]. Questions should promote various skills and dispositions, such as evaluation, and include words from Bloom's taxonomy [43]. Socratic questioning, as well as questions not entailing a right or wrong answer, can set the stage for students to explore different beliefs or opinions through investigation and comparison [44].

Even though the strategies mentioned above, alongside others (i.e., dialogue and mentorship), may significantly support students in developing CT skills, it should be noted that they are no panacea unless they are blended and combined during the teaching process. Nevertheless, teaching CT does not include a 'magic recipe', but some of the above can function as the first steps of this process [30] (p. 303). Moreover, students also stressed that carrying out research and practicing in real contexts can contribute to fostering their CT [45].

The third challenge has recently arisen under the lens of the Framework Theory. It refers to the conceptualization of CT, and the ideas students might have before any explicit instruction of CT in HEI. Students' prior beliefs about CT might activate different processes when they face problems, dilemmas or case studies included in CT instruction. For instance, a student who asserts that CT is about asking questions and criticizing will activate different cognitive processes in comparison to a student who does believe that CT is just thinking or to another who believes that CT is a necessary vehicle to understand science. Therefore, the academic staff should be aware of these beliefs and implement different teaching approaches that could trigger cognitive processes to help students overcome the barriers from the initial ideas for CT.

### 1.4. The Current Study

Current psychology approaches relate the conceptual changes that occur when students learn scientific concepts to the changes in their belief systems, and particularly with the ideas students have about various aspects of their studies before they confront the scientific theories. According to the Framework theory approach of conceptual change [7], novices in a field conceptualize a scientific concept differently than experts. Novices construct intuitive theories for phenomena and processes related to them based on the everyday experience that validates and confirms them. In many scientific domains, these theories are rational, similar to theories formulated in the past, and perceived as scientific for their time. The current scientific theories might enrich them, or they might define the

core concepts radically differently [46]. Deep learning of a scientific concept, then, requires radical changes in the novice's initial conceptualizations, and not mere enrichment of their initial concepts.

Understanding the difference between the novices' and experts' theories on a topic is crucial for designing and implementing the instruction [6]. Students' ideas about a concept could be seen as a constantly evolving structure activated to interpret incoming information [47] (p. 230). Without considering how the novices' 'theories' and the experts' theories are connected, instruction of the scientific concept might result in constructing misconceptions, namely, erroneous interpretations of the scientific concept. Misconceptions are considered as novices' efforts to synthesize and integrate aspects of the scientific view into their prior 'theories' [7]. Conceptual change is a slow and cumbersome process, with recent evidence suggesting that initial theories co-exist with scientific theories even in experts [48].

Under this rationale, understanding students' conceptualization of CT when they enter the HEI and their ideas of how it can be acquired becomes crucial for designing and developing evidence-based interventions or courses. For instance, if students hold beliefs identical to scientific concepts, the course's emphasis could be on implementing the concept in various contexts. If students have similar but incomplete concepts (e.g., they lack only some aspects of the concept), the instructor should facilitate the acquisition of scientific knowledge through the enrichment of the initial concepts. On the contrary, if students hold misconceptions about the concept, the instructor should first deconstruct them by employing techniques such as cognitive dissonance before presenting the scientific concept. Finally, if students show ignorance about the scientific concept, the instructor might start by introducing the concept from scratch. All these cases might be present among students in the same class, and the instructor should implement differentiated instruction [49] to tackle the various types of students' concepts.

Drawing on the above, it becomes vital for Higher education to consider the first-year students' conceptualizations of CT to design effective educational interventions that could prepare students to meet the labor market expectations. To the best of our knowledge, there is no such study that examines how students conceptualize CT at the very beginning of their studies. The current exploratory study aimed to map students' ideas about CT and how CT skills can be acquired. Based on the Framework theory approach [7], we could assume that students, when entering the HEI, have constructed an understanding of the concept. This concept might result from the combination of their studies in previous levels of education and their everyday experience, which may be identical or different from the definition academics utilize for CT. It is crucial to capture the possible alternative conceptions first-year students might bring to HEI. The level of confidence in the endorsements of particular aspects of a concept has been used as an additional measurement to differentiate and categorize possible erroneous conceptions as those arising due to either a lack of prior knowledge or the existence of entrenched alternative conceptions [50]. Measurements of the certainty and confidence individuals have about their endorsement of a statement can enlighten the researchers about whether (a) the respondents have the scientific idea (i.e., they answer correctly and are confident); (b) they have a blurred scientific conception (i.e., they answer correctly but they are not confident); (c) they have an alternative conception or misconception (i.e., they answer incorrectly and are confident); (d) they have an initial blurred incorrect concept and are confident; (e) they are ignorant of the issue (i.e., they denote ignorance with confidence); or (f) they show ignorance of the aspect under discussion without being fully aware of whether they know or don't know about it.

## 2. Materials and Methods

### 2.1. Participants

The total sample of 243 (N = 186 female, N = 51 male, N = 3 other, N = 3 I prefer not to say) participants were first-year students studying Social Sciences (i.e., Education and Psychology) at a University in Northwestern Greece. CT is neither explicitly addressed

nor taught as a stand-alone subject in secondary education in Greece. Therefore, we could argue that when students enter HEI, they lack theoretical or scientific knowledge about CT, but they only have an initial conceptualization of CT due to the implicit teaching and practice of CT skills during secondary education and their everyday experience.

### 2.2. Procedure

This study received ethical approval from the Research Ethics Committee of the University (Reg. No.: 21-2023/25-10-2022). In the current study, we followed convenience sampling as a non-probability sampling approach [51]. In particular, two of the present study's authors were first-year student instructors. Students were invited to participate in the survey without additional extrinsic motivation (e.g., extra credits or tokens). Data collection took place at the beginning of the winter 2022–2023 semester. Participants were given a QR code for an online survey developed by the authors using Google Forms. Participation was anonymous. At the beginning of the survey, students were informed of the aim and objectives of the study as well as the policy regarding their data identification. Further, participants were informed that answering all questions included in the survey was compulsory. The research was presented as a mapping study of students' ideas regarding CT. Thus, to avoid any priming effect, the terms 'misconceptions', 'initial ideas', 'alternative ideas', 'alternative concepts', 'non-scientific statements', or 'scientific statements' were not mentioned in the information provided to the students. The average time for completing the questionnaire was 15 min.

### 2.3. Measurements

In order to explore students' ideas about CT, the authors constructed a new research instrument. The authors created a pool of 30 statements regarding the conceptualization and assessment of CT and administrated them to 63 students. The mean scores for ten out of the 30 statements were found at the chance level ($ps > 0.001$). These items were excluded from the version of the instrument administered in the main survey. Therefore, the survey included the remaining 20 statements about CT; particularly, 11 non-scientific and nine scientific statements. However, the statements varied in content. Half of them (i.e., 10 out of the 20 statements) concerned the conceptualization of CT, namely how CT is defined. The remaining ten statements referred to students' ideas of how CT might be acquired. For the statements regarding the conceptualization of CT, five were scientific statements that were either exactly quoted from the existing literature (i.e., statements 1, 5 and 10) or rephrased from common CT definitions (i.e., statements 3 and 6). The non-scientific statements about the conceptualization of CT were (i) incomplete definitions of CT (i.e., statement 8), (ii) statements challenging the two-dimensional nature of CT (i.e., statements 4 and 7), and (iii) common mistakes regarding the nature of CT (i.e., statements 2 and 9). All the non-scientific statements regarding the conceptualization of CT were inspired or inferred by the respective literature [e.g., statement 2, inspired from 10].

Moreover, the remaining ten statements in the survey referred to students' ideas regarding how CT might be acquired. Four statements were scientific, and six were non-scientific. The content of the scientific statements focused on (i) the instructional approaches and strategies that could promote CT (i.e., statements 11 and 17), (ii) the transferability of CT (i.e., statement 13), (iii) and the mental processes learners follow when engaging with CT (i.e., statement 16). Four statements were extracted from CT-related literature, but two of them were quotations from Halpern's previous work on CT (i.e., statement 16 [21] and statement 13 [15]). The non-scientific statements about how CT might be acquired referred to (i) the misconceptions that CT is a general learning process and a matter of developing general thinking skills (i.e., statements 12 and 15), (ii) the debate that CT can be nurtured through instruction (i.e., statements 14, 18 and 19), and (iii) the disciplines where CT can be enhanced (i.e., statement 20). Three non-scientific statements were quoted from the respective CT literature (i.e., statements 14, 18 and 19), while the rest of the non-scientific statements were inspired by or inferred from the literature. Participants indicated their

level of agreement with each statement using a 5-point Likert scale (1: totally disagree, 5: totally agree). The non-scientific statements were reversed; thus, higher scores (i.e., 4 and 5 on the scale) denote participants' agreement with the scientific conceptions, and lower scores (i.e., 1 and 2 on the scale) their disagreement with the scientific conceptions. The order of the scientific and the non-scientific statements was randomized.

Moreover, for each statement, participants had to express their level of certainty (1: just guessing, 6: extremely certain) on the Certainty Response Index (CRI) e.g., [50]. Surveys integrating the CRI have been employed mainly in science education research to map students' misconceptions e.g., [50,51]. The hypothesis is that the correct (scientific) response, accompanied by high (i.e., 5 and 6 on the scale) certainty about the answer, indicates (scientific) knowledge. Low confidence (i.e., 1 and 2 on the scale), regardless of the answer provided, denotes ignorance. Finally, wrong responses accompanied by high levels of certainty (i.e., 5 and 6 on the scale) display entrenched misconceptions [50].

In the current study, participants were categorized into six groups according to their endorsements and confidence level. In the first group, participants endorsing a fact about CT with high certainty (i.e., Scientific/Confident) were classified. This group of participants was perceived as having scientific knowledge regarding CT. The second group consisted of participants who endorsed a scientific statement regarding CT with low certainty (i.e., Scientific/Unconfident), namely participants with a blurred scientific conception. The third group of participants included answers denoting neither agreement nor disagreement with the statements in combination with high certainty (i.e., Ignorance/Confident). Hence, this group of participants was perceived as ignorant of the under-discussion aspect of CT. The fourth group was composed of participants denoting neither agreement nor disagreement with the statements in combination with low certainty (i.e., Ignorance/Unconfident), thus participants were classified as ignorant of the aspect but not fully aware of what they don't know or know. The fifth group encompassed participants endorsing a non-scientific statement about CT with low certainty (i.e., Non-Scientific/Unconfident). Participants, in this case, were considered as having preconceptions, namely, an initial blurred incorrect concept of CT. Finally, the last group consisted of participants who endorsed a non-scientific statement about CT with high certainty (i.e., Non-Scientific/Confident). Participants in this group were considered as having a misconception/alternative conception, namely an erroneous conceptualization of CT.

### 3. Results

*3.1. Preliminary Analysis of the Instrument*

See Tables 1 and 2 below.

**Table 1.** Mean scores, standard deviations, certainty level and comparison of participants' endorsement regarding the statements about the conceptualization of CT against chance level.

| | Statements | M | SD | *t* | M Certainty | SD Certainty |
|---|---|---|---|---|---|---|
| 1 | *Critical Thinking entails awareness of one's own thinking and reflection on the thinking of self and others as an object of cognition* [24] (p. 270). | 4.07 | 0.85 | *28.713 \** | *4.14* | *1.12* |
| 2 | Critical thinking is a clear concept with a clear definition [14,52]. | 4.02 | 0.88 | *9.920 \** | *3.76* | *1.13* |
| 3 | *The ideal critical thinker can be characterized for both her or his cognitive skills and also for her or his habits of mind* [20]. | 3.73 | 0.83 | *23.086 \** | *3.82* | *1.15* |

**Table 1.** *Cont.*

| | Statements | M | SD | t | M Certainty | SD Certainty |
|---|---|---|---|---|---|---|
| 4 | A person that is disposed towards critical thinking is engaged in assessing and validating information [20]. | 3.60 | 0.96 | *17.905 ** | *3.95* | *1.09* |
| 5 | *Critical thinking is a vehicle for comparing assertions to reality and determining their truth or falsehood* [22] (p. 311). | 3.58 | 0.97 | *17.229 ** | *3.84* | *1.07* |
| 6 | *Critical Thinking has been related to analytic thinking processes, which are purposeful, self-regulatory, conscious and effortful* [17]. | 3.40 | 0.99 | *14.126 ** | *3.67* | *1.11* |
| 7 | Errors in thinking often occur not because people cannot think critically, but because they are unwilling to [48]. | 3.38 | 1.12 | *23.086 ** | *4.08* | *1.02* |
| 8 | Critical Thinking is the ability to engage in challenging discussions and to analyze and interpret information [53]. | 3.29 | 1.16 | *10.506 ** | *4.29* | *0.9* |
| 9 | Someone thinks critically when engaging in criticism, namely when judging or questioning the merits and faults of some content or facts [54]. | 3.28 | 1.07 | *11.272 ** | *3.86* | *0.88* |
| 10 | *Critical thinking is valued as a vehicle that promotes sound assertions and enhances understanding* [55] (p. 364). | 3.11 | 0.98 | *26.829 ** | *4.35* | *1.02* |

Note: Scientific statements are in italics. * $p < 0.001$.

**Table 2.** Mean scores, standard deviations, certainty level and comparison of participants' endorsement regarding the statements about how CT might be acquired against chance level.

| | Statements | M | SD | t | M Certainty | SD Certainty |
|---|---|---|---|---|---|---|
| 11 | *Various instructional approaches can benefit my students towards the development of their Critical Thinking skills, such as problem-based learning, dilemma's discussions, and case studies* [31,35,37]. | 4.14 | 0.74 | 34.446 * | 4.31 | 1.06 |
| 12 | Teaching critical thinking is primarily a matter of developing thinking skills [48]. | 3.99 | 0.88 | 26.385 * | 4.28 | 1.09 |

**Table 2.** *Cont.*

| | Statements | M | SD | *t* | M Certainty | SD Certainty |
|---|---|---|---|---|---|---|
| 13 | *It is vital to direct a person's learning so that the skills of Critical Thinking are learned in a way that will facilitate their recall in novel situations [19].* | 3.99 | 0.9 | 25.841 * | 4.25 | 1.11 |
| 14 | Asking challenging questions and presenting opposite views on a topic seem appropriate teaching strategies to exploit in order to promote Critical Thinking [56]. | 3.80 | 1.03 | 19.721 * | 4.20 | 1.03 |
| 15 | Critical thinking involves generic operations that can be learned by following a set of steps, apart from any particular knowledge domain, and can be transferred to or applied in different contexts [53]. | 3.69 | 0.98 | 18.812 * | 3.97 | 1.12 |
| 16 | *When engaging in Critical Thinking, a person needs to monitor their thinking process, check whether progress is being made toward an appropriate goal, ensure accuracy and make decisions about the use of time and mental effort [25].* | 3.67 | 0.98 | 18.494 * | 3.92 | 1.17 |
| 17 | *There are specific types of questions that I can use to trigger students' different critical thinking skills and critical thinking dispositions [25,42,44,57].* | 3.50 | 1.02 | 15.159 * | 3.87 | 1.12 |
| 18 | Participating in (group) discussions or brainstorming activities suffice to foster the development of my Critical Thinking [30]. | 3.26 | 1.11 | 10.605 * | 4.21 | 0.99 |
| 19 | A person cannot develop their Critical Thinking because there are no appropriate instructional approaches or teaching strategies that can promote the development of Critical Thinking [30]. | 3.20 | 1.36 | 8.029 * | 4.45 | 0.90 |
| 20 | Critical thinking can only be taught in those disciplines where explicit problem-solving methodologies can be applied, e.g., medicine [30]. | 1.56 | 0.75 | -19.485 * | 4.71 | 0.98 |

Note: Scientific statements are in italics. * $p < 0.001$.

To examine the validity of the statements included in the survey, the statements were compared against chance level (defined at 2.5). Clear trends were revealed ($p < 0.001$) for all statements identified as scientific or non-scientific. Tables 1 and 2 present the mean scores and standard deviations regarding participants' endorsement and their level of certainty for each statement included in the survey.

### 3.2. The Conceptualization of Critical Thinking

The data were analyzed using SPSS v22. Each statement was categorized into one of the six groups (i.e., Scientific/Confidence, Scientific/Unconfidence, Ignorance/Confidence, Ignorance/Unconfidence, Non-scientific/Unconfidence, Non-scientific/Confidence; for the classification process, see: Measurements). Undergraduate students' ideas and their scientific knowledge, initial blurred scientific concepts, ignorance, initial blurred non-scientific concepts and alternative conceptions regarding the conceptualization of CT are presented in Figure 1. For a comprehensive presentation of the results, we first present the analysis of the scientific statements (see Figure 1, statements 1 to 5) followed by the non-scientific statements (see Figure 1, statements 6 to 10).

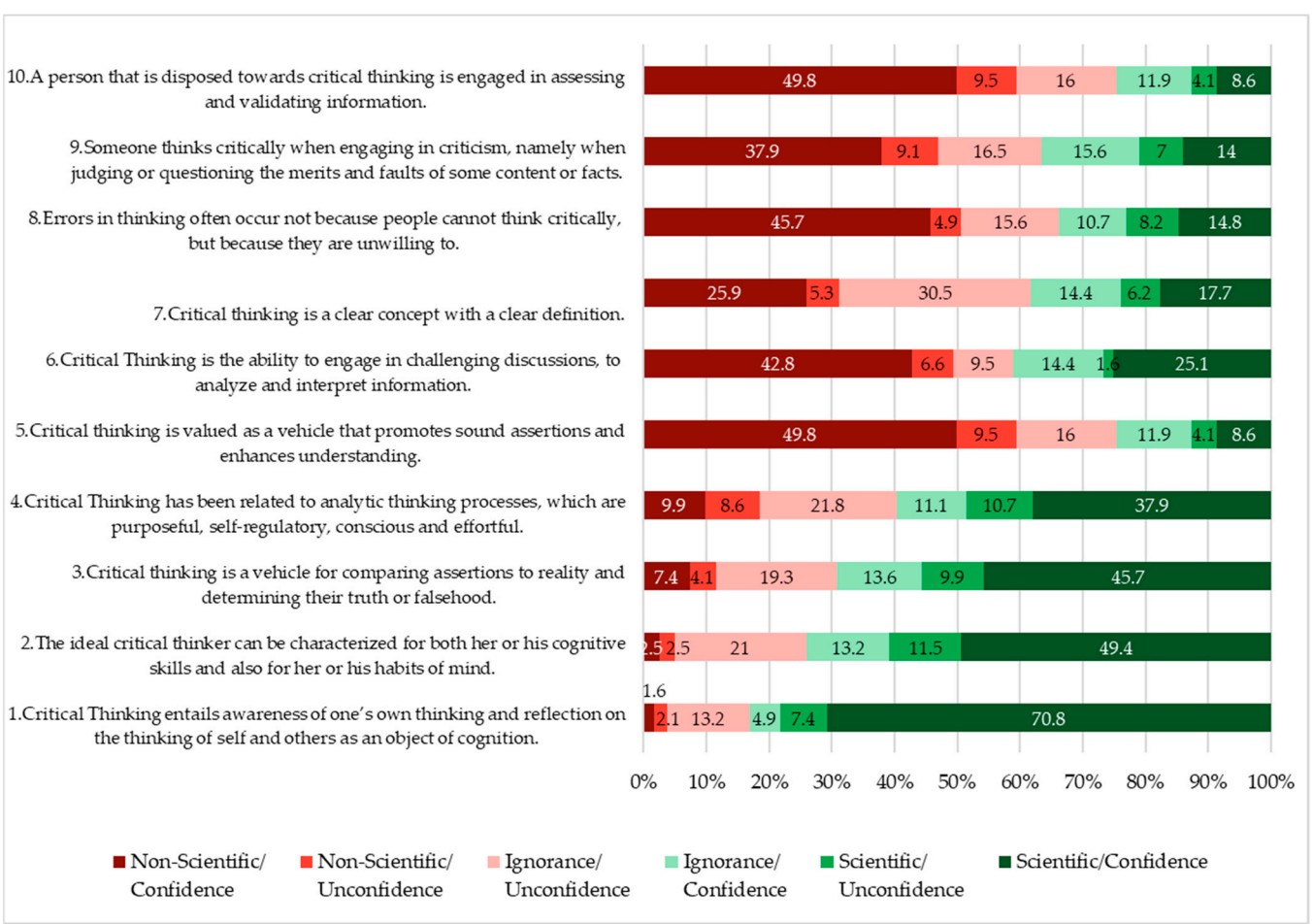

**Figure 1.** Students' conceptualization of CT.

Overall, the analysis of the scientific statements (i.e., statements 1 to 5) revealed that in four out of five statements, most students endorsed the scientific ideas with confidence. In particular, scientific statement 1 (i.e., CT entails awareness of thinking and reflection) was endorsed by most students (70.8%) with confidence, indicating that students' conceptualization reflects the scientific view. Further, 7.4% of the participants endorsed the scientific statement, but they were not certain about their endorsement, indicating that

they hold some initial blurred conceptualization of CT for this aspect. Less than 20% of the participants displayed ignorance. A small percentage of the participants (i.e., 2.1%) rejected the scientific statement, but they were not certain about their endorsement, and only 1.6% of participants were found to have an alternative conception.

However, the subsequent three scientific statements, namely statement 2 (i.e., CT thinking is related to skills and habits of mind), 3 (i.e., CT is the means for comparing assertions to reality), and 4 (i.e., CT involves metacognitive monitoring), revealed participants' bewilderment. Less than half of the students endorsed the scientific view with confidence, a percentage between 49.4% and 37.9%. Less than 20% of participants in all three statements endorsed the items, but they were not certain, indicating that they held some initial blurred ideas about the conceptualization of CT. Moreover, more than 30% of the students indicated their lack of knowledge; namely, they were ignorant, being either confident or unconfident, and a very small percentage of participants between 2.5% and 9.9%, had an alternative conception.

Nevertheless, in the last scientific statement, namely, statement 5 (i.e., CT is related to argumentation and enhancement of understanding), only a few (8.6%) presented ideas aligned with the scientific view and were confident or unconfident (4.1%), denoting that they hold some initial blurred idea about the relationship between CT and argumentation. However, almost half (49.8%) of the students' ideas were misconceptions; a few (9.5%) had non-scientific initial blurred ideas, while almost 30% of the participants indicated their lack of knowledge.

The analysis of the non-scientific statements (see Figure 1, statements 6–0) revealed that in four out of five statements, a high percentage of students had ideas that were alternative conceptions to the scientific view. Specifically, the non-scientific statements 6 (i.e., CT is engagement in challenging discussions, analysis and interpretation), 8 (i.e., unwillingness to think results in errors), 9 (i.e., CT is engagement in criticism) and 10 (i.e., CT dispositions are assessment and validation of information) were faultily endorsed by the majority of the participants. In particular, only a few participants, a percentage that ranked between 25.1% and 8.6%, had ideas in alignment with the scientific view, while a small percentage of students, between 16.1% and 8.2%, either endorsed or rejected the scientific view without confidence.

However, a considerable number of students endorsed the non-scientific statements, and they were confident in their endorsement (i.e., 42.8%, 45.7%, 37.9%, and 49.8%, for 6, 8, 9 and 10, respectively), revealing alternative conceptions regarding the conceptualization of CT. A surprising observation is that only a few (17.7%) of the students rejected the non-scientific statement 7 (i.e., CT is a clear concept) with confidence, and fewer (6.2%) rejected the non-scientific statement without confidence. One out of four students (26%) considered alternative conceptions. Finally, overall, up to 45% of the participants displayed a lack of knowledge.

### 3.3. How Critical Thinking Might Be Acquired

Following the same methodology we followed above for the conceptualization of CT, students' answers about each statement were classified into one of the six groups. The undergraduate students' ideas and their scientific knowledge, initial blurred scientific concepts, ignorance, initial blurred non-scientific concepts and alternative ideas regarding how CT might be acquired, are presented in Figure 2. For a comprehensive presentation of the results, we first present the analysis of the scientific statements (see Figure 2, statements 1 to 4) following by the non-scientific statements (see Figure 2, statements 5 to 10).

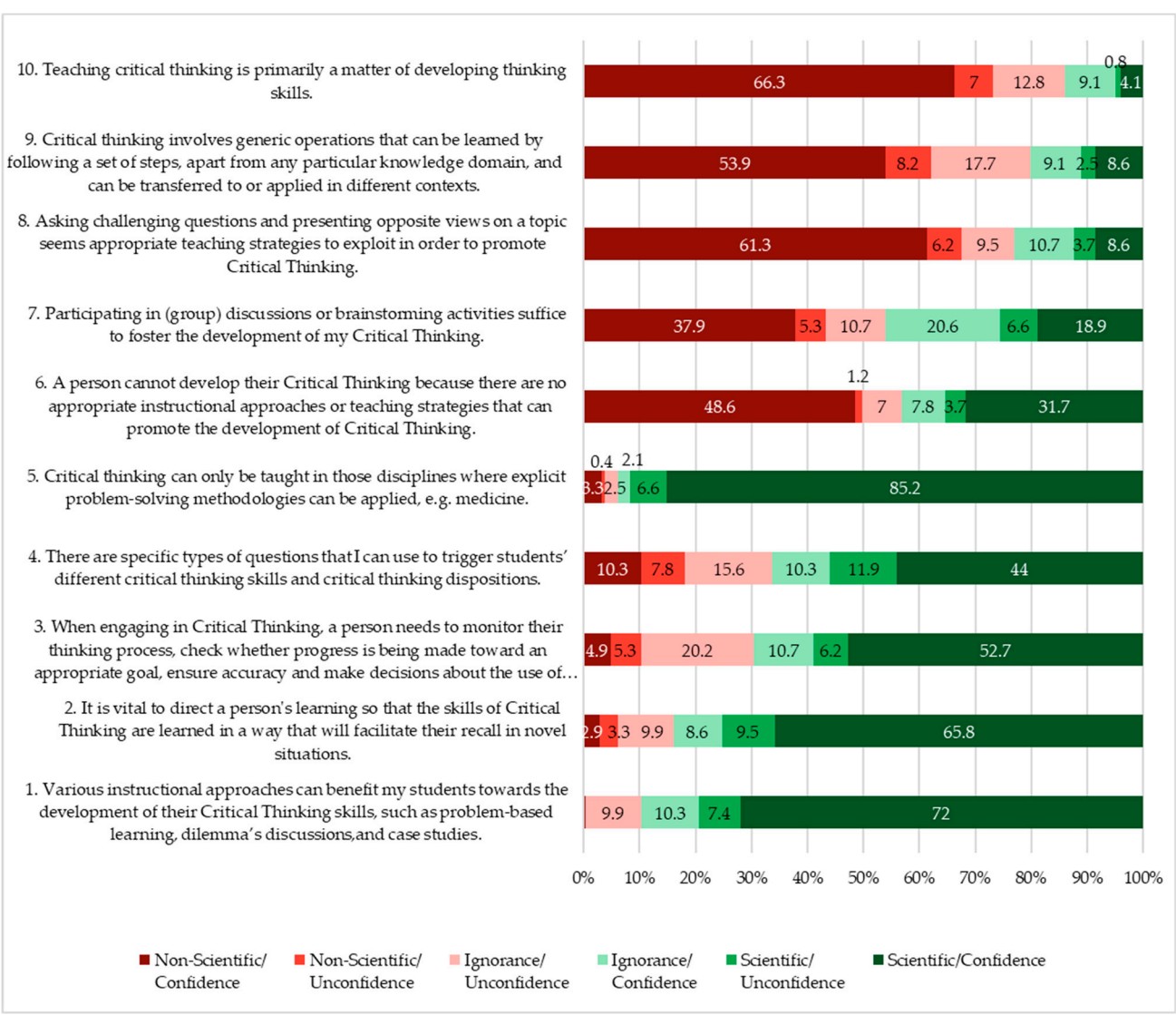

**Figure 2.** Students' conceptualization of CT.

Figure 2 shows that three out of the four scientific statements, namely statements 1 (i.e., various instructional approaches promote CT), 2 (i.e., transferability of CT skills), and 3 (i.e., CT requires self-regulation), indicated that the majority of participants held ideas about the enhancement of CT in alignment with the scientific view (the percentages ranging between 72% and 52.7%), while few of the participants, ranking between 10.7% and 7.4%, endorsed the scientific statements without being confident. However, a relatively high percentage of participants, between 30.8% and 18.5%, expressed their ignorance about the validity of the statements. Furthermore, a small percentage (i.e., 3.3% for statement 2 and 5.3% for statement 3) had non-scientific initial blurred ideas. Finally, a very small percentage of the participants' ideas (i.e., 0.4%, 2.9% and 4.9% for statements 1, 2, and 3, respectively), could be considered alternative conceptions.

At the same time, for the remaining scientific statement that is statement 4 (i.e., specific questions nurture CT), less than half (44%) of the participants' ideas were in line with the scientific view, while almost 10.3% of students endorsed the scientific statement without being confident. Moreover, 7.8% seemed to have an initial blurred idea as to whether certain questions could nurture CT. Surprisingly, more than a quarter of the participants indicated their lack of knowledge when considering the particular statement, while a small percentage of (10.3%) declined to accept the validity of the scientific statement and were certain about the incorrectness of the statement.

As far as the non-scientific statements are concerned, it was striking that only one out of the six non-scientific statements, namely statement 5 (i.e., CT can be taught in specific disciplines), was rejected by the majority of the participants (i.e., 85.2%), while few students (6.6%) endorsed the scientific statement without being confident. Additionally, 0.4% rejected the scientific statement without being certain. Thus, the majority of students declined to accept the idea that CT should be taught in specific courses. Finally, a small percentage of students expressed their lack of knowledge (i.e., 4.6%) or endorsed the statement with certainty (3.3%).

Further, statements 6 (i.e., CT cannot be taught), 8 (i.e., asking questions and argumentation promotes CT), 9 (i.e., CT is domain-general), and 10 (i.e., CT is about the instruction of skills) revealed that participants endorsed the non-scientific statements with certainty (i.e., 48.6%, 61.3%, 53.9% and 66.3% for 6, 8, 9, and 10 respectively) indicating they had misconceptions. Only in statement 6 (i.e., CT cannot be taught), did 31.7% of the participants reject the non-scientific assertion with confidence, indicating that their ideas were in accordance with the scientific view. The respective percentage regarding participants' scientific ideas about statements 8 (i.e., asking questions and argumentation promotes CT), 9 (i.e., CT is domain-general), and 10 (i.e., CT is about the instruction of skills) were between 8.6% and 4.1%, while only a small percentage of participants endorsed the non-scientific assertions but without certainty (i.e., 3.7%, 3.7%, 2.5% and 0.8% for the 6, 8, 9, and 10 respectively). Also, a small percentage rejected the non-scientific statements while being uncertain about their judgment (i.e., 1.2%, 6.2%, 8.2% and 7% for 6, 8, 9, and 10 respectively). Finally, a number of participants (i.e., 14.7%, 20.2%, 26.8% and 21.9% for the 6, 8, 9, and 10, respectively) avoided endorsing or rejecting the assertions, indicating their lack of knowledge about the statement.

Lastly, regarding the non-scientific statement 7 (i.e., group discussions and brainstorming promote CT) participants correctly rejected the statement with confidence (18.9%). A few participants endorsed the non-scientific statement without confidence (6.6%). Additionally, few participants rejected the non-scientific statement while being unconfident (5.3%). However, 37.9% of the participants endorsed the assertion with confidence, indicating entrenched misconceptions. Finally, a considerable percentage of participants (31.3%) expressed ignorance, either with high (20.6%) or low (10.7%) certainty.

## 4. Discussion

The present study was exploratory and aimed to map first-year social sciences students' conceptualization of CT and their ideas regarding how CT might be acquired. We followed the Framework theory approach [7], according to which we assumed that, when entering Higher education, students have already formed a conceptualization of CT. Their conceptualization might result from their studies in previous levels of education as well as their everyday experience, and it might be identical to or different from the scientific view on CT. To meet the aim of the current study, we collected 20 statements found in the literature. These reflect the current scientific conceptualization or non-scientific ideas which research showed that individuals, even scientists, adopted for CT conceptualization and acquisition in the past. Participants endorsed or rejected the statements and expressed their certainty in their judgment. This method helped us to classify participants across six categories that reflect different conceptualization of CT or how CT could be acquired. The acknowledgement of the six categories among undergraduate students leads HEI teaching staff to design and implement different teaching approaches through the lens of differentiated instruction [49].

### 4.1. The Conceptualization of Critical Thinking

Our findings indicated that students conceptualize CT as akin to scientists, and they are certain about the validity of their conceptions only for two CT aspects examined here. Students recognized that (i) CT is a two-dimensional construct involving skills and dispositions and (ii) metacognitive thinking processes are engaged in CT. Indeed, these

conceptions align with experts who conceptualize CT under the skills and disposition paradigm e.g., [18,20,26–28] and with those who see metacognition as an essential component of CT e.g., [21,24]. A small percentage of first-year students were found to endorse the scientific and reject the non-scientific statements intuitively, without being confident about the validity of their decisions. These results might indicate that although not explicitly taught, some students seem to have been familiarized with some aspects of CT at previous education levels. Thus, they exhibit an initial blurred conceptualization of CT.

Beyond these two aspects, many students (49.8%–37.9%) had well-entrenched misconceptions about some assertions concerning the conceptualization of CT (see Box 1). That is, they endorsed the non-scientific or rejected the scientific statements, and they were very confident about the validity of their decisions. The most frequent misconceptions found among first-year students were about the relationship between CT and argumentation, CT and enhancement of understanding (49.8%) and whether a disposed of individual towards CT is engaged in assessing and validating information (49.8%). Moreover, four out of ten students held a misconception connecting the errors in thinking with the individuals' will think critically (45.7%) as well as understanding CT as only engagement in challenging discussions, analysis, and interpretation (42.8%), or only engagement in criticism (37.9%). These misconceptions show that students hold a (mis)conceptualization of CT that even some experts in the field of CT have expressed in the past [53]. This can be interpreted if we consider the lack of explicit teaching or reference to the conceptualization of CT in syllabi or textbooks in previous levels of education, but one can find only some aspects of CT immersed in learning and teaching [4]. This kind of familiarization leads students to integrate only some aspects of the scientific view into their prior ideas which are covered by the scientific concepts as well as creating a synthetic model for the concept, namely the misconception [7]. Students' misconceptions as these arose from the statements used in the questionnaire are generally described in Box 1.

**Box 1.** Students' misconceptions about the conceptualization of CT and how CT might be acquired.

> 1. Critical thinking is not about advancing correct claims and enhancing understanding.
> 2. A person disposed to think critically mainly focuses on assessing information for validity.
> 3. Everyone who participates in challenging discussions and is able to analyze and interpret information is a good critical thinker.
> 4. Everyone can think critically but some do not want to.
> 5. Critical thinking is the ability to criticize facts or arguments as well as to identify advantages and disadvantages regarding an issue.
> 6. Critical thinking cannot be taught in a class.
> 7. Asking challenging questions in a classroom and bringing controversial views regarding a topic is enough to promote students' critical thinking.
> 8. Critical thinking involves a set of steps of general operations that can be learned and applied in any context.
> 9. Teaching critical thinking is primarily a matter of developing thinking skills.

Nevertheless, a high percentage of participants were ignorant of some aspects of the conceptualization of CT; they avoided endorsing or rejecting the statements. More than 30% of students ignored the two-dimensional construct of CT that CT can be the means for comparing assertions to reality and recognizing their truthfulness or falsehood, that CT is not a clear concept, and whether CT is a simple engagement in criticism. This finding is in line with previous work from Bezanilla, Galindo-Domínguez, and Poblete [58], who illustrated that HEI teachers noted that their students lack previous conceptualization about CT when entering tertiary education.

*4.2. How Critical Thinking Might Be Acquired*

As far as how CT might be acquired is concerned, our findings indicated that students have some understanding on the issue. In particular, most students recognized that

various instructional approaches, such as problem-based learning and case studies, could activate critical thinking skills and dispositions and should be designed to facilitate the transferability of CT in novel situations. In addition, they acknowledged that CT requires self-regulation and that instruction of CT can be implemented in many disciplines aside from those implementing problem-solving methodologies like medicine. These aspects of CT are recognized by scholars in the field of CT and are widely accepted as they appear in the relevant literature [19,30,31,35,37,56].

However, a small group of students had initial blurred ideas regarding how CT might be acquired. This group of participants was identified in all statements included in the survey examining students' ideas about how CT might be acquired. Previous research suggests that undergraduate social science students' recognition of the instructional approaches that promote CT could be attributed to students' knowledge acquisition about these instructional approaches and experience in these instructional approaches during their studies [45]. Hence, our finding regarding the small percentage of students with initial blurred ideas is anticipated, considering that the first-year students engaged in our study completed the survey at the beginning of the semester having received no instruction.

Nevertheless, our results underlined that a large percentage (66.3%–53.9%) of students having misconceptions about how CT might be acquired (see Box 1). Specifically, more than half (66.3%) of the students held the idea that CT is related to the acquisition of skills, can be promoted simply by asking questions or engaging in argumentation (61.3%), is domain-general (53.9%), and there is an inability to nurture CT through instruction (48.6%). Previous research has revealed that even experts have misconceptions regarding concepts such as the domain specificity, the domain-general nature of CT or whether CT is a matter of developing thinking skills [53].

Finally, it should be mentioned that about a quarter of the students denoted their ignorance regarding the steps one can follow to develop CT regardless of the domain (26.8%) as well as the specific questions that nurture CT (25.9%). This percentage was even higher concerning the assertion that group discussions and brainstorming promote CT (31.3%) as well as the statement that CT requires self-regulation (30.9%). These results are consistent with the existing literature, which suggests that not only students [59] but also instructors in higher education [60,61] lack knowledge about how CT might be promoted.

### 4.3. Implications for Instruction

From an instructional point of view, our results provide evidence that: (a) students do have a limited understanding regarding the conceptualization of CT and how CT might be developed, and (b) they also have various ideas on these concepts, which should be taken into consideration when it comes to courses aiming to help students in the direction of CT acquisition. Hence, systematic instruction about CT across the curricula and different courses that are part of a program in order to treat limited understanding of CT could prove to be essential for students in higher education [62]. However, instruction ignoring students' different initial conceptualizations would prove to be inefficient in terms of conceptual change [6,7].

Instruction addressing the topic to students who have already grasped the scientific conceptualization could focus on the implementation and transferability of their understanding to various authentic problems related to the course. Instruction addressing the topic to students with scientific but blurred ideas could focus on the consolidation and enrichment of their initial conceptualization. For this group of students, the conceptualization of CT could be achieved through explicit instruction [63–65]. Knowledge acquisition regarding the conceptualization of CT and how CT might be developed could be promoted through the design of inquiry-based learning environments [30,66–68]. Inquiry-based learning environments provide students with the opportunity to deal with authentic problems aligned with the professional contexts [30] and to engage in dialogue with their peers, which could enhance their metaconceptual awareness about the coherence and explanatory power of their concepts [69].

An instruction to students holding misconceptions about CT demands more sophisticated teaching approaches. Previous work has shown that inquiry-based learning environments could prove beneficial in tackling students' misconceptions [70,71]. Instructors could also employ cognitive conflict in their teaching (see Limón [72] for a discussion on cognitive conflict) before presenting the scientific concept to their students. An additional teaching strategy triggering cognitive conflict is the refutation text (see Tippett [73] for a review). In particular, a refutation text explicitly states a misconception and then refutes it with the presentation of the scientific idea. Thus, students are more likely to recognize that their prior idea is incorrect or inadequate, and they need to reconstruct it in order to comply with the scientific idea.

Finally, instruction aiming at students lacking knowledge of CT concepts could follow two different approaches depending on students' level of certainty regarding their ignorance. In the first case, namely ignorant students who are confident, instruction could start from scratch. Hence, explicit instruction on the concepts of CT and instruction fostering the acquisition of scientific knowledge could be employed. Therefore, ignorant and confident students could be treated similarly to students having initial blurred ideas about CT. In the second case, we consider it essential to engage ignorant and unconfident students in an intervention that focuses not only on cognitive aspects but also on motivational factors. Demetriou and colleagues [74] showed that self-awareness mediates the general intelligence factor and the general factor of personality, highlighting that self-awareness is important for the feelings of self-worth, confidence, self-efficacy and motivation to engage in mental processing. Drawing on the above, we assume that ignorant and unconfident students were most likely aware of the accuracy or inaccuracy of their ideas on CT. Still, they lacked confidence in their cognitive abilities and motivation to engage in thinking processes. Thus, they preferred to denote their ignorance while partaking in the survey. Therefore, at an instructional level, we suggest that it would be essential for instructors to design learning environments that satisfy students' basic psychological needs for self-competence and autonomy, promoting in return their motivation to engage in learning and instruction [75]. At the beginning of the intervention, instructors could implement guided inquiry activities in order to scaffold students to feel more competent and confident. Once students feel confident, the instructor can focus on instruction promoting CT. Nevertheless, it is unclear whether students' ideas are either accurate or inaccurate. Thus, instructors could treat them similarly to students with initial blurred ideas, focusing on explicit instruction of CT and the implementation of activities fostering the acquisition of scientific knowledge concerning CT.

### 4.4. Limitations

Although our results show clear trends in students' responses, this study has some limitations. First, we developed a research instrument based on the current, influential and highly accepted literature in the field of CT for both scientific as well as alternative ideas. However, the statements derived from this literature review refer only to some aspects of CT and the teaching methods that seem to promote CT. Additionally, the study's participants are students prepared to attend the school of social sciences, namely Education and Psychology. It is possible for students following different types of preparation (e.g., focusing on science, argumentation, etc.) to exhibit different levels of understanding about CT. Finally, the current study was exploratory, and the structure of the undergraduate students' ideas, namely whether they form a coherent belief system or they are fragmented, was not examined.

### 4.5. Future Research

Future research should explore in-depth students' misconceptions about CT, either by enriching our pool of alternative ideas or by using a different research methodology. Moreover, future research could examine CT understanding in different study disciplines as well as during the entrance into the labor market. Further, the structure (coherent system

belief vs. fragmented) of undergraduate students' ideas on CT could be examined. This is an essential step in the process of conceptual change. In particular, rendering learners aware of their conceptions and making their ideas the point of reflection might foster their conceptual change on CT and improve the quality of teaching interventions at a higher education level or at a professional development level. To that end, as existing research suggests, not only students but also instructors tend to have misconceptions that they pass on to their students [76]. Thus, it would be interesting to explore further and address instructors' ideas or beliefs on CT, which in turn might influence their practices in learning and instruction. Finally, future research could examine how students perceive the impact of different teaching approaches (e.g., problem-based learning, dilemma discussions, case studies) on their perceived ability to think critically.

## 5. Conclusions

CT has been recognized as one of the most important traits of the 21st century. The current study suggested a sophisticated approach (borrowed from the Framework Theories for conceptual change [7]) for approaching the conceptualization of CT and how CT might be acquired. We studied first-year social sciences' undergraduates and suggested six categories of students that have distinct conceptualizations of CT. In particular, the current study clarified for the first time a number of misconceptions that first-year students have constructed through their prior education (see Box 1). Moreover, classifying the students into one of the six categories is helpful for HEI instructors. We suggested different teaching approaches be applied in the class for students of different types to benefit from constructing a coherent conceptualization of CT.

The accurate conceptualization of CT is only the first step towards preparing students on CT for the labor market needs. Future employees should also be prepared to solve problems, self-correct and self-regulate, minimize decision-making mistakes, act responsibly [77], and become proactive in times of crisis [78]. It is not sure, however, that HEI and the Labor Market have a common understanding of all these aspects. For instance, labor market might see the responsibility of the employees in terms of profit-making goals, while HEI in terms of social responsibility (e.g., enhancing the trust in science for sensitive public issues such as pollution) [78,79] or adopting a client-centered approach [80]. Future research should enlighten more about how HEI and labor market conceptualize these aspects, revealing the values they expect will guide employees' critical decisions.

**Author Contributions:** Conceptualization, D.P., P.C. and T.G.; Methodology, D.P., and P.C.; Formal Analysis, D.P., P.C. and T.G.; Investigation, D.P., T.G. and A.L.; Resources, D.P., P.C., T.G. and A.L.; Data Curation, P.C.; Writing—Original Draft Preparation, P.C.; T.G. and A.L.; Writing—Review & Editing, D.P.; P.C.; T.G., and A.L.; Visualization, P.C.; Supervision, D.P.; Project Administration, D.P.; Funding Acquisition, D.P. All authors have read and agreed to the published version of the manuscript. All authors agree to be personally accountable for their contributions and for ensuring that questions related to the accuracy or integrity of any part of the work, even ones in which the authors were not personally involved, are appropriately investigated, resolved, and documented in the literature.

**Funding:** This research was supported by the "Critical Thinking for Successful Jobs—Think4Jobs" Project, with grant number 2020-1-EL01-KA203078797, funded by the European Commission/EACEA, through the ERASMUS + Programme. The European Commission support for the production of this publication does not constitute an endorsement of the contents, which reflect the views only of the authors, and the Commission cannot be held responsible for any use, which may be made of the information contained therein.

**Institutional Review Board Statement:** The study was conducted in accordance with the Declaration of Helsinki. In addition, the Ethics Committee of the University of Western Macedonia approved the study (Reg. No.: 21-2023/25-10-2022).

**Informed Consent Statement:** Informed consent was obtained from all participants involved in the study.

**Data Availability Statement:** The raw data supporting the conclusions of this article will be made available by the authors, without undue reservation.

**Conflicts of Interest:** The authors declare no conflict of interest.

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
