# Peer review of "Undergraduate Students’ Conceptualization of Critical Thinking and Their Ideas for Critical Thinking Acquisition"

_education, doi:10.3390/educsci13040416_

Round 1

Reviewer 1 Report

At line 88-98 the paper is not clear. 

Conclusion are a little bit superficially developed, it would be useful a deeper consideration. The working world is not prepared to accept Critical thinking with enthusiasm.

The reference part should be reinforced and empowered with citation about the application of CT to a sustainable and cooperative approach as:

Barrena Martinez, J., Lopèz Fernandez, M., and Romero Fernandez, P., M. (2016). Corporate social responsibility: Evolution through institutional and stakeholder perspectives. European Journal of Management and Business Economics, 25 (1), 8-14. Doi: 10.1016/j.redee.2015.11.002

Borchorst, N. G., McPhail, B., Smith, K. L., Ferenbok, J., & Clement, A. (2012). Bridging identity gaps - supporting identity performance in citizen service encounters. Computer Supported Cooperative Work, 21(6), 555-590. doi:10.1007/s10606-012-9163-5

Pascucci, T. , Cardella, M. G. , Hernàndez-Sanchez, B. & Sanchez-Garcìa, J. C. (2022). Systematic Review of Socio-Emotional Values within Organizations. Frontiers in Psychology, 12, 738203. https://doi.org/10.3389/fpsyg.2021.738203

Author Response

Response to Reviewer #1

At line 88-98 the paper is not clear. 

The text was modified in order to present clearly the ideas:

[Research evidence supports that CT skills and dispositions can be taught by employing specific teaching strategies in all scientific fields and educational levels [26, 27]. Having an agreement that CT skills and dispositions should be taught, scholars are confronted with three new challenges. The first challenge has been thoroughly discussed and refers to CT transferability from one domain to another. Transferability is considered a process through which critical thinkers can activate specific CT skills needed to tackle a new situation. To this end, they can anatomize the issue, identify its ‘blind spots’ and at the same time overcome superficial or naïve approaches [15]. It should be noted, though, that transferability can take place when do-main and or/background knowledge regarding the issue is also present [28].]

Conclusion are a little bit superficially developed, it would be useful a deeper consideration. The working world is not prepared to accept Critical thinking with enthusiasm.

We agree with the Reviewer that the working world is not prepared to accept Critical Thinking with enthusiasm. However, this aspect might be an RQ of another study. Nevertheless, the Reviewer triggered a discussion among the authors for particular topics that the labor market and HEI might have different priorities. Based on this comment and discussion, we reformulated the Conclusions, always keeping in mind that this aspect is beyond the scope of this manuscript. 

[CT has been recognized as one of the most important traits of the 21st century. The current study suggested a sophisticated approach (borrowed from the Framework Theories for conceptual change [7]) for approaching the conceptualization of CT and how CT might be acquired. We studied first-year social sciences’ undergraduates and suggested six categories of students that have distinct conceptualizations of CT. In particular, the current study clarified for the first time a number of misconceptions that first-year students have constructed through their prior education (see Box 1). Moreover, classifying the students into one of the six categories is helpful for HEI instructors. We suggested different teaching approaches be applied in the class for students of different types to benefit from constructing a coherent conceptualization of CT.

The accurate conceptualization of CT is only the first step towards preparing students on CT for the labor market needs. Future employees should also be prepared to solve problems, self-correct and self-regulate, minimize decision-making mistakes, act responsibly [77], and become proactive in times of crisis [78]. It is not sure, however, that HEI and the Labor Market have a common understanding of all these aspects. For instance, labor market might see the responsibility of the employees in terms of profit-making goals, while HEI in terms of social responsibility (e.g., enhancing the trust in science for sensitive public issues such as pollution) [78, 79] or adopting a client-centered approach [80]. Future research should enlighten more about how HEI and labor market conceptualize these aspects, revealing the values they expect will guide employees' critical decisions.]

The reference part should be reinforced and empowered with citation about the application of CT to a sustainable and cooperative approach as:

Barrena Martinez, J., Lopèz Fernandez, M., and Romero Fernandez, P., M. (2016). Corporate social responsibility: Evolution through institutional and stakeholder perspectives. European Journal of Management and Business Economics, 25 (1), 8-14. Doi: 10.1016/j.redee.2015.11.002

 Borchorst, N. G., McPhail, B., Smith, K. L., Ferenbok, J., & Clement, A. (2012). Bridging identity gaps - supporting identity performance in citizen service encounters. Computer Supported Cooperative Work, 21(6), 555-590. doi:10.1007/s10606-012-9163-5

Pascucci, T. , Cardella, M. G. , Hernàndez-Sanchez, B. & Sanchez-Garcìa, J. C. (2022). Systematic Review of Socio-Emotional Values within Organizations. Frontiers in Psychology, 12, 738203. https://doi.org/10.3389/fpsyg.2021.738203

We found the suggested literature relevant, and we added them to our manuscript [79, 80, 78]

Reviewer 2 Report

Dear Authors,

Thank you for submitting your manuscript which I really enjoyed reading it! 

Your study is organized around a coherent set of questions and includes classic studies related to the questions guiding the study. The Discussion part takes the form of a logical argument that concludes with a clear rationale for additional research.

Please find comments and questions below:

- Line 34-40 requires references

- Was the questionnaire piloted before being introduced to students?

- Were all the questions asked in the questionnaire compulsory to answer?

- It might be interesting to study students' conception regarding the benefits of CT activities such as PBL in the future. 

The introduction clearly addresses the importance of CT conceptualization and why this needs to considered before designing teaching activities that involve CT. The research main question was supported with the results evidencing some misconceptions or insufficient conceptualization of CT. This is a gap and rather overlooked in HEI including CT perception assessment for instructors that is also mentioned as the future research within this paper.    The methodology and associated data analyses seemed appropriate. Table captions could be more descriptive. It is suggested to define the statistical values used within the tables.     The conclusion is consistent with the evidence and arguments presented and addresses the study question. References are appropriate!

Thank you!

Author Response

Response to Reviewer #2

- Line 34-40 requires references

The following references were added to further support the content of lines 34-40:

  • Vosniadou, S. The Framework Theory Approach to the Problem of Conceptual Change. International handbook of re-search on conceptual change, 2; Vosniadou, S. Ed.; Routledge: New York, 2013, pp. 23-42
  • Duro, E.; Elander, J.; Maratos, F. A.; Stupple, E. J.; Aubeeluck, A. In search of critical thinking in psychology: an exploration of student and lecturer understandings in higher education. Psychol Learn Teach 2013, 12 (3), 275-281. http://dx.doi.org/10.2304/plat.2013.12.3.275
  • Lloyd, M.; Bahr, N. Thinking Critically about Critical Thinking in Higher Education. JoSoTL, 2010, 4 (2). https://doi.org/10.20429/ijsotl.2010.040209

- Was the questionnaire piloted before being introduced to students?

We thank Reviewer #2 for this comment. We agree that this information is necessary and we added this information in the 2.3 Section. 

In order to explore students’ ideas about CT, the authors constructed a new research instrument. The authors created a pool of 30 statements regarding the conceptualization and assessment of CT and administrated them to 63 students. The mean scores for ten out of the 30 statements were found at the chance level (ps > .001). These items were excluded from the version of the instrument administered in the main survey. Therefore, the survey included the remaining 20 statements about CT.

- Were all the questions asked in the questionnaire compulsory to answer?

A sentence was added to clarify this:

Further, participants were informed that answering all questions included in the survey was compulsory. (Lines 215-216).

- It might be interesting to study students' conception regarding the benefits of CT activities such as PBL in the future. 

We agree with Reviewer#2 for this comment. We perceived this comment as a question for future research. Thus, we added a sentence to Section 4.5 - Future Research (Lines 597-599).

Finally, future research could examine how students perceive the impact of different teaching approaches (e.g., problem-based learning, dilemma discussions, case studies, etc.) on their perceived ability to think critically.

The introduction clearly addresses the importance of CT conceptualization and why this needs to considered before designing teaching activities that involve CT. The research main question was supported with the results evidencing some misconceptions or insufficient conceptualization of CT. This is a gap and rather overlooked in HEI including CT perception assessment for instructors that is also mentioned as the future research within this paper. The methodology and associated data analyses seemed appropriate.

We thank Reviewer#2 for the positive evaluation of these parts.

Table captions could be more descriptive.

Table 1 % 2 captions were described in more detail.

It is suggested to define the statistical values used within the tables.

The statistical values depicted in Tables 1 & 2 are in all cases smaller than 0.001. We searched the Journal Guidelines for how to present the statistical values in Tables and found no relevant suggestions. Thus, due to space restrictions (i.e., an additional column would lead us to minimise the font size), we deemed it more appropriate to signal the value with a note that corresponds to all statements’ comparison against chance level.  

The conclusion is consistent with the evidence and arguments presented and addresses the study question.

References are appropriate!

We thank Reviewer#2 for the positive comments on our conclusion and References.

Reviewer 3 Report

Lines 99-122 . Why doesn't the author mention cooperative learning as one of the most powerful teaching approaches for promoting critical thinking skills and dispositions?

Lines 202-202 . Does what is stated seem to contradict what is said in lines 436 to 438?

Author Response

Response to Reviewer #3

Lines 99-122 . Why doesn't the author mention cooperative learning as one of the most powerful teaching approaches for promoting critical thinking skills and dispositions?

The text was slightly changed and references were added to include cooperative learning as one approach that promotes CT skills and dispositions:

  • Loh, R. C. Y.; Ang, C. S. Unravelling Cooperative Learning in Higher Education: A Review of Re-search. RESSAT, 2020, 5 (2), 22-39. doi.org/10.46303/ressat.05.02.2
  • Zhang, J.; Chen, B.; The effect of cooperative learning on critical thinking of nursing students in clinical practicum: A quasi-experimental study. J Prof Nurs, 2021, 37 (1), 177-183. https://doi.org/10.1016/j.profnurs.2020.05.008

Lines 202-202 . Does what is stated seem to contradict what is said in lines 436 to 438?

The text in both cases was modified and stated more precisely to avoid any contradictory wording.

The modification at lines 201-204:

Therefore, we could argue that when students enter HEI, they lack theoretical or scientific knowledge about CT, but they only have an initial conceptualization of CT due to the implicit teaching and practice of CT skills during secondary education and their everyday experience.

The modification at lines 447-449:

These results might indicate that although not explicitly taught, some students seem to have been familiarized with some aspects of CT at previous education levels. Thus, they exhibit an initial blurred conceptualization of CT.
